# Chitosan Nanoparticle-Encapsulated *Cordyceps militaris* Grown on Germinated *Rhynchosia nulubilis* Reduces Type II Alveolar Epithelial Cell Apoptosis in PM_2.5_-Induced Lung Injury

**DOI:** 10.3390/ijms26031105

**Published:** 2025-01-27

**Authors:** Hyo-Min Kim, Jong-Heon Kim, Byung-Jin Park, Hye-Jin Park

**Affiliations:** Department of Food Science and Biotechnology, College of BioNano Technology, Gachon University, Seongnam-si 13120, Republic of Korea; hyomin0401@gmail.com (H.-M.K.); kjhss960909@gmail.com (J.-H.K.); qudwls4733@gmail.com (B.-J.P.)

**Keywords:** particulate matter, lung injury, *Cordyceps militaris* grown on germinated *Rhynchosia nulubilis* encapsulated in chitosan nanoparticles (GCN), oxidative stress, ROS-driven apoptosis, inflammation-driven apoptosis

## Abstract

Chitosan nanoparticles (CNPs) were synthesized in this study to enhance the limited bioactivity and stability of *Cordyceps militaris* grown on germinated *Rhynchosia nulubilis* (GRC) and effectively deliver it to target tissues. Under optimized conditions, stable encapsulation of GRC was achieved by setting the chitosan (CHI)-to-tripolyphosphate (TPP) ratio to 4:1 and adjusting the pH of TPP to 2, resulting in a zeta potential of +22.77 mV, which indicated excellent stability. As the concentration of GRC increased, the encapsulation efficiency decreased, whereas the loading efficiency increased. Fourier-transform infrared (FT-IR) spectroscopy revealed shifts in the amide I and II bands of CHI from 1659 and 1578 to 1639 cm⁻^1^, indicating hydrogen bonding and successful encapsulation of GRC encapsulated with CNPs (GCN). X-ray diffraction (XRD) examination revealed the transition of the nanoparticles from a crystalline to an amorphous state, further confirming successful encapsulation. In vivo experiments demonstrated that GCN treatment significantly reduced lung injury scores in fine particulate matter (PM_2.5_)-exposed mice (*p* < 0.05) and alleviated lung epithelial barrier damage by restoring the decreased expression of occludin protein (*p* < 0.05). In addition, GCN decreased the PM_2.5_-induced upregulation of *MMP-9* and *COL1A1* mRNA expression levels, preventing extracellular matrix (ECM) degradation and collagen accumulation (*p* < 0.05). GCN exhibited antioxidant effects by reducing the mRNA expression of nitric oxide synthase (*iNOS*) and enhancing both the protein and mRNA expression of superoxide dismutase (*SOD-1*) caused by PM_2.5_, thereby alleviating oxidative stress (*p* < 0.05). In A549 cells, GCN significantly reduced PM_2.5_-induced reactive oxygen species (ROS) production compared with GRC (*p* < 0.05), with enhanced intracellular uptake confirmed using fluorescence microscopy (*p* < 0.05). In conclusion, GCN effectively alleviated PM_2.5_-induced lung damage by attenuating oxidative stress, suppressing apoptosis, and preserving the lung epithelial barrier integrity. These results emphasize its potential as a therapeutic candidate for preventing and treating the lung diseases associated with PM_2.5_ exposure.

## 1. Introduction

Exposure to fine particulate matter (PM) leads to adverse health effects. PM flows into the body through breathing, causing severe damage to the respiratory system [1]. The World Health Organization (WHO) announced that nine out of ten individuals breathe high concentrations of air pollutants and estimated that seven million premature deaths are caused by air pollution every year [2]. The number of PM-related patients in the Middle East and Eastern Asia is consistently increasing [3,4]. There were 8536 death tolls in 2017 due to PM_2.5_ (PM_2.5_, particulate matter with a diameter 2.5 μm or smaller) in Saudi Arabia, a part of the Middle East. It was estimated that 54% of PM-induced mortality was attributable to ischemic heart disease, 16% to lower respiratory infections, 12% to stroke, 7% to chronic obstructive pulmonary disease (COPD), 7% to type II diabetes, and 4% to lung cancer. PM-induced diseases in East Asia show similar patterns, with most studies indicating that lung diseases are the most severely affected health issue by PM exposure [5,6]. Furthermore, recent studies have demonstrated that PM_2.5_ exposure strongly increased the mortality rate related to coronavirus 2019 (COVID-19). The ongoing impact of COVID-19, resulting from the highly contagious SARS-CoV-2 virus, is exacerbated by PM exposure and has proven fatal to humanity because of the high mortality and morbidity associated with chronic lung diseases [7,8]. PM_2.5_ can penetrate deep into the lung and form deposits in the alveoli owing to its large surface area and relatively small particle size compared to PM_10_. This leads to altered lung functionality and the ability to enter the bloodstream by crossing the alveolar–capillary barrier [9,10,11]. PM exposure triggers various respiratory diseases, including asthma, lung cancer, COPD, and pulmonary fibrosis (PF). These conditions arise primarily through oxidative damage, inflammation, cell cycle dysregulation, macrophage dysfunction, and disruption of intracellular calcium homeostasis [1]. Among them, inflammation and oxidative stress are major triggers for PM-induced respiratory diseases [12,13]. PM commonly contains polycyclic aromatic hydrocarbons (PAHs), metals, elemental carbon, and inorganic ions. The constituents of PM induce excessive reactive oxygen species (ROS) generation, causing inflammation and oxidative stress. This affects the airways and lungs, resulting in either intrinsic or extrinsic apoptosis [14,15].

*Cordyceps militaris* has traditionally been used as a folk remedy in various Asian countries, including China. *C. militaris* contains useful chemical constituents such as cordycepin, polysaccharides, ergosterol, adenosine, and mannitol [16]. Polysaccharides, which are abundant in *C. militaris*, possess various biological effects including anti-inflammatory, antioxidant, immunomodulation, and antithrombotic activities [17,18]. These active compounds, like cordycepin and polysaccharides in *C. militaris*, are well known for preventing respiratory diseases due to their anti-inflammatory and antioxidant properties [19,20,21]. However, the production yield of *C. militaris* in nature is extremely low due to its requirement for specific hosts and rigorous growth conditions. The commercial use of natural *C. militaris* is restricted due to its low extraction yield and high expense [16].

To address these issues, artificial cultivation methods for the large-scale production of *C. militaris* have been explored [22]. Our group selected germinated *Rhynchosia nulubilis* as a culture medium for *C. militaris* cultivation, instead of dead insects, using patented technology [23,24,25].

The black soybean (*Glycine max* L. Merr), referred to as *R. nulubilis* (Yak-Kong), is a legume species native to East Asia that is widely utilized in traditional oriental medicine [26]. It is rich in proteins, with a well-balanced profile of amino acids, isoflavones, anthocyanins, sterols, phytic acid, saponins, and phenolic compounds. These phytochemicals can prevent various chronic diseases and exert health-promoting benefits due to their anti-inflammatory and antioxidant properties [27,28,29]. Furthermore, primary isoflavones, including genistein, daidzein, and glycitein, increase antioxidant enzyme activity and lower oxidative stress levels, preventing PM-induced lung diseases [30,31]. Our previous study showed that germinated *R. nulubilis* suppressed PM-induced type II alveolar epithelial (A549) cell death owing to its antioxidant effects [32].

Our previous analysis revealed that *C. militaris* cultivated on germinated soybean (GSC) extracts had higher concentrations of phytochemicals, including flavonoids and total phenolics, than germinated soybeans (GSs). Furthermore, GSC demonstrated significantly higher antioxidant activity than GSs [24]. Therefore, the increased productivity and enriched bioactive components of *C. militaris* grown on germinated *R. nulubilis* (GRC) could overcome the limitations of natural *C. militaris*, resulting in enhanced biological activity. Despite its enhanced physiological activity, the practical use of GRC is limited when it is administered orally. Poorly soluble compounds in GRC have low bioavailability and are highly excreted from the body, requiring a high dose and repeated oral administration [33,34]. For instance, phenolic compounds in GRC exhibit poor bioavailability due to their low absorption and instability in the gastrointestinal (GI) tract caused by insufficient gastric residence time, low permeability, and solubility [33]. Water-soluble components, such as polysaccharides and isoflavones from GRC, have limited efficacy when administered orally because their poor epithelial permeability and inadequate absorption through the mucosa hinder their antioxidant potential in combating oxidative stress [35]. Furthermore, sensitive, physiologically active compounds such as cordycepin and isoflavones exhibit low stability and storability when exposed to light, heat, pH, and oxygen [36,37].

Improving the bioavailability and stability of active substances is crucial for strengthening the role of GRC as a natural food and medicinal product in the food industry. To address these challenges, we used nanoparticle drug delivery systems to encapsulate GRC. These systems are designed to incorporate active substances into nanocarriers, improve their absorption in the GI tract, and facilitate targeted delivery to specific sites or cells, such as enterocytes or M cells. Targeting depends on factors such as pH, enzyme and receptor reactivity, and electrostatic interactions [38,39,40]. Using this technique, the poor solubility and instability of GRC can be overcome, thereby increasing the absorption rate and retention time of poorly soluble or sensitive compounds in the body [41].

Among the various nanoparticle-based systems, chitosan nanoparticles (CNPs) have emerged as promising nanocarriers due to their ability to improve the biological activity and oral bioavailability of numerous phytochemicals [42,43]. In addition to being non-toxic and cost-effective, CNPs are biodegradable and biocompatible, and they are synthesized from the natural polymer chitosan (CHI). These nanoparticles exhibit high permeability across biological membranes and are susceptible to enzymatic hydrolysis, making them ideal for diverse applications. Their popularity has grown significantly in recent years [44,45,46,47]. Encapsulating GRC in CNPs (referred to as GCN) protects its bioactive compounds from degradation and inactivation in the GI tract caused by biological defense mechanisms such as low pH and digestive enzymes. The sustained release of these components, combined with enhanced solubility and stability, enables the effective absorption of the bioactive compounds of GRC into target tissues [48,49,50,51,52].

Our study demonstrated that GCN exhibits improved bioavailability in mouse small intestinal tissues [53]. The cationic polysaccharide CHI in CNPs interacts easily with negatively charged cell membranes through electrostatic interactions, allowing CNPs to cross cellular membranes via active endocytosis [54,55,56]. This mechanism significantly improves the cellular uptake of water-soluble components in GCN, enhancing its ROS scavenging activity. Consequently, GCN exhibits stronger antioxidant properties, effectively countering PM-induced oxidative stress compared to unencapsulated GRC. Moreover, the increased mucoadhesion of GCN facilitates mucosal absorption, promoting the efficient release of GRC extracts into the lung tissues and systemic circulation. This targeted delivery enables the bioactive compounds of GRC to exert their effects more effectively in tissues, such as the lungs [49,52].

In addition, encapsulating GRC in the form of freeze-dried GCN powder enhances its stability, making it easier to handle and store than unencapsulated GRC [57,58,59]. Previously, we confirmed the anti-inflammatory and antioxidant effects of GCN in inflammatory lung disease triggered by PM_2.5_ exposure [53]. However, the potential of GCN in mitigating PM_2.5_-induced pulmonary dysfunction remains to be fully explored. In the present study, we examined whether GCN could attenuate apoptosis and oxidative stress induced by PM_2.5_ exposure in a murine model and A549 cells.

## 2. Results

### 2.1. Entrapment Efficiency and Loading Efficiency of GCN

In our previous study, we tested various CHI–tripolyphosphate (TPP) ratios and pH levels to determine the optimal conditions for GCN synthesis (Figure 1A) [53]. In that study, we found that GRC CHI nanoparticles exhibited nanoscale sizes (160, 146, and 188 nm) at GRC concentrations of 2, 4, and 8 mg/mL, respectively, and showed zeta potentials of +35.6, +34.41, and +30.68 mV, as measured by nanoparticle tracking analysis (NTA). Among these, CNPs synthesized with a CHI:TPP ratio of 4:1 and a TPP pH of 2 were selected as the optimal conditions for GRC encapsulation. This selection was based on the zeta potential value, indicating high stability compared to other formulations [60].

The encapsulation efficiency (EE%) and loading efficiency (LE%) of the CNPs were evaluated at varying GRC concentrations (2, 4, and 8 mg/mL). The EE% values were 39.6 ± 0.94%, 36.7 ± 1.35%, and 31.4 ± 1.32% for GRC concentrations of 2, 4, and 8 mg/mL, respectively. The LE% values were 1.59 ± 0.48%, 4.39 ± 0.32%, and 7.62 ± 0.33% for the same concentrations (Figure 1B). These results indicate a decreasing trend in the EE% with increasing GRC concentration, whereas the LE% increased up to 8 mg/mL. This could be due to the higher concentration of active compounds (such as proteins, polysaccharides, and polyphenols) in GRC, which reduces the availability of CHI to entrap GRC. Consistent findings were observed by Soltanzadeh et al. and Lee et al., with higher extract concentrations causing a decrease in the available CHI moiety for encapsulating the extract, resulting in reduced EE% [61,62].

**Figure 1 ijms-26-01105-f001:**
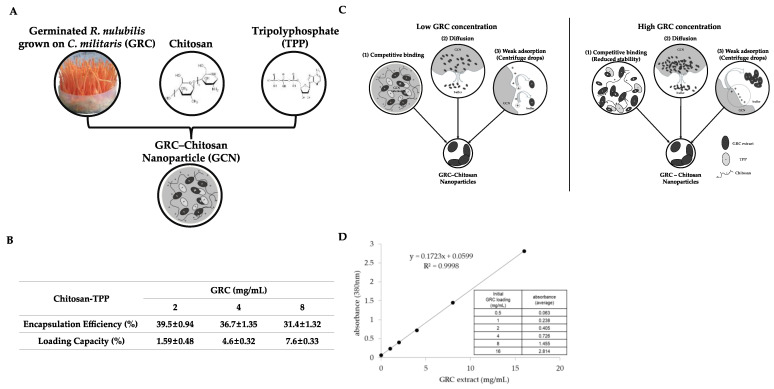
Encapsulation and release characteristics of GRC encapsulated in CHI-TPP nanoparticles. (**A**) Encapsulation of GRC encapsulated in CHI-TPP nanoparticles, (**B**) encapsulation efficiency (EE%) and loading capacity (LC%) of GRC-loaded CNPs, (**C**) factors affecting the EE%, and (**D**) calibration curve of the initial GRC extract using a UV spectrophotometer (380 nm) and its regression value [63,64,65,66,67].

### 2.2. Structural Characterization of GCN Using FT-IR and XRD

To verify the successful entrapment of the GRC extract in nanoparticles, FT-IR spectroscopy and XRD were used to analyze the interactions between CNPs and GRC. FT-IR spectroscopy revealed the characteristic peaks of pure CHI at 3445 cm^−1^ (N-H and O-H stretching vibrations) and 1659 and 1578 cm^−1^ (amide I-II bands) [68].

In the CNP spectrum data without the GRC extract, the amino group peaks from CHI shifted from 1659 and 1578 cm^−1^ to 1636 cm^−1^ owing to ionic bonding with TPP, and the P=O peak of TPP was observed at 1208 cm^−1^ [68,69]. In the GCN spectrum, the O-H and N-H stretching peaks shifted from 3410 cm^−1^ to 3378 cm^−1^ and appeared broader, indicating hydrogen bonding interactions. The disappearance of the peaks at 1659 cm^−1^ and 1578 cm^−1^ (amide I and II of CHI) and the modified peak at 1639 cm^−1^ suggest interactions with GRC compounds.

The FT-IR spectrum of the GRC extract revealed a characteristic peak at 1644 cm^−1^ (C=O or amide I band), with broad absorption peaks at 3451 cm^−1^ and 1401 cm^−1^ due to hydroxyl groups [20,23,70,71,72,73]. Additionally, a significant peak at 1419 cm^−1^, corresponding to phenolic compounds from GRC, was observed in the GCN spectrum (Figure 2A) [69,72,73].

Furthermore, the opaque color change in GCN and the disappearance of the crystalline peak in the XRD analysis indicate a change in the CHI structure upon interaction with TPP, revealing a transition to amorphous nanoparticles (Figure 2B) [74].

The in vitro release analysis revealed that after 2 h, the release percentage of GRC from GCN at pH 7.4 and 2.2 was 21.3 ± 1.8% and 23.7 ± 2.7%, respectively, whereas, at pH 5.5, it was 14.8 ± 1.4% (*p* < 0.05) (Figure 2C). Drug release was significantly higher at pH 2.2 owing to the partial dissolution/swelling of CNPs in a strongly acidic environment [75,76], whereas higher diffusion was observed at pH 7.4 due to CHI deprotonation [77,78].

### 2.3. Improved Bioavailability of GCN

To evaluate the enhanced cellular uptake and utilization of GRC encapsulated in CHI nanoparticles, we treated A549 cells with FITC-GRC or FITC-GCN 200 μg/mL. Cellular uptake was observed after 2 or 4 h using fluorescence microscopy (Figure 3A,B). Enhanced cellular uptake of FITC-GCN was observed compared with FITC-GRC at both 2 and 4 h (*p* < 0.05).

### 2.4. GCN Alleviates PM_2.5_-Induced Pulmonary Damage in Mice

To examine the ameliorative effects of GCN on pulmonary damage induced by PM_2.5_, hematoxylin and eosin (H&E) staining was carried out and lung injury scores were measured. The lungs treated with PM_2.5_ exhibited alveolar wall thickening, hemorrhaging, and inflammatory cell infiltration, whereas these impairments were less pronounced with GCN pretreatment. Furthermore, compared to the GCN pretreatment, the PM_2.5_-treated group exhibited a distinct expansion of the alveolar space due to the destruction of the alveolar wall (Figure 4A). Next, we compared the lung injury scores of GCN- and PM_2.5_-treated mice with a semiquantitative histopathology scoring system. The lung injury score of the PM_2.5_-treated group was notably higher than that of the control group (*p* < 0.05). The score was significantly decreased in the GCN-treated group compared to the PM_2.5_-treated group (*p* < 0.05) (Figure 4B). These findings show that the lung tissue destruction and injury induced by PM_2.5_ were reduced by GCN. Furthermore, we assessed whether the GCN and GRC treatments had toxic effects on normal mice by monitoring the changes in body weight. Alterations in body weight are commonly considered reliable markers to evaluate the toxicity of test samples in vivo [79]. As shown in Figure 4C, no significant alterations were observed in the body weights of the mice orally treated with 2 g/kg GCN or 2 g/kg GRC compared to the phosphate-buffered saline (PBS)-treated mice over two weeks.

### 2.5. GCN Effect on Lung Barrier Integrity Disruption and Collagen Synthesis Induced by PM_2.5_ In Vivo

Next, we examined the expression level of tight junction (TJ) proteins and the levels of *MMP-9* and *COL1A1* mRNA to elucidate the alleviating effects of GCN on the lung epithelial barrier dysfunction triggered by PM_2.5_. Occludin, a key regulator of TJ protein function and stability, is associated with lung epithelial barrier integrity [80]. We found that the occludin protein expression level was significantly decreased within the PM_2.5_-treated group. As depicted in Figure 5, the occludin protein expression was significantly elevated after GCN treatment compared to that in the GRC and PM_2.5_ treatments (*p* < 0.05). Moreover, mRNA expression levels of *MMP-9* and *COL1A1* were significantly increased in the PM_2.5_-treated group, whereas they were significantly decreased in the GCN-treated group (*p* < 0.05) (Figure 5B,C). In addition, the mRNA expression levels of *COL1A1* and *MMP-9* were significantly lower with GCN treatment compared to GRC (*p* < 0.05). These data suggested that GCN attenuated lung epithelial barrier dysfunction and PF in the PM_2.5_-treated mice.

### 2.6. GCN Suppresses PM_2.5_-Induced A549 Cell Death

We investigated whether GCN affected PM_2.5_-induced A549 cell death. GCN significantly reduced PM_2.5_-induced A549 cell death in a dose-dependent manner between 50 and 200 µg/mL (*p* < 0.05) (Figure 6A). Notably, GCN with a concentration of 200 μg/mL suppressed PM_2.5_-induced A549 cell death when compared with GRC with a concentration of 200 µg/mL (*p* < 0.05). Therefore, 200 μg/mL GCN was used for further experiments. Accordingly, the effective dose of 50% (ED50) of GRC in A549 cells was 231.45 ± 99.16 μg/mL, whereas the ED50 of GCN was 123.69 ± 5.83 μg/mL, approximately twice that of GRC. In addition, the lethal dose of 50% (LD50) of GRC was 735.75 ± 136.53 μg/mL, whereas the LD50 of GCN was 2327.9 ± 568.15 μg/mL, an approximately threefold increase compared to GRC (Figure 6B,C). The therapeutic index (TI) was calculated to compare the relative safety of GCN and GRC in terms of overdose risk, revealing a TI of 18.82 ± 4.7 for GCN and 3.18 ± 1.33 for GRC. This represents an approximately sixfold increase in the TI for GCN compared to that for GRC, suggesting a statistically significant increase in the safety and efficacy of GCN in inhibiting PM_2.5_-induced A549 cell death (independent sample *t*-test; *p* < 0.01) [81].

### 2.7. GCN Reduced PM_2.5_-Induced ROS Generation

Oxidative stress is a main factor of PM-induced lung injury. We investigated whether GCN can block oxidative stress in PM_2.5_-treated lungs and A549 cells. For this purpose, the nitric oxide synthase (*iNOS)* and *SOD-1* mRNA expressions were measured in PM_2.5-_ and GCN-treated murine lungs. Compared to the control group, the *iNOS* mRNA expression level was significantly higher in the PM_2.5_-treated group, but this increase was diminished by GCN treatment in the murine lungs (*p* < 0.05) (Figure 7A). The GCN-treated group showed notably higher mRNA expression of *SOD-1* compared to the PM_2.5_-treated group (*p* < 0.05). Moreover, the *SOD-1* mRNA expression was notably elevated with the GCN pretreatment compared to GRC. To assess the effect of GCN on SOD-1 protein expression in PM-treated murine lung tissue, immunohistochemical (IHC) staining was performed for SOD-1. As shown in Figure 7B, GCN treatment markedly upregulated SOD-1 protein expression in the murine lungs compared with the PM_2.5_-treated group. According to previous studies, PM generates ROS, which are the major contributors to cell death and inflammation [82]. We evaluated the radical scavenging activity of GCN against PM_2.5_-induced ROS production in A549 cells. 2-7-dichlorodihydrofluorescein diacetate (DCF-DA) was employed to assess intracellular ROS levels. The elevated fluorescence intensity revealed that PM_2.5_ induced a 1.47 ± 0.1-fold increase in intracellular ROS generation compared to the control (*p* < 0.05). In addition, compared to GRC in intracellular ROS generation, GCN caused a significant 0.86 ± 0.0-fold reduction (*p* < 0.05) (Figure 7C).

### 2.8. GCN Suppresses Both PM_2.5_-Induced Intrinsic and Extrinsic Apoptosis Pathways

To elucidate the inhibitory effect of GCN on PM_2.5_-induced lung epithelial barrier apoptosis, apoptosis-related protein expression levels were measured in both lungs and A549 cells. The protein expression levels of PM_2.5_-induced p53 and cleaved caspase-3/9/8 decreased following GCN pretreatment compared with the PM_2.5_-treated group (*p* < 0.05). As shown in Figure 8A,B, GCN treatment significantly upregulated the expression of the anti-apoptotic protein Bcl-2 and downregulated the expression of the apoptotic protein Bax (*p* < 0.05). Correspondingly, the protein levels of Bax/Bcl-2 reduced after GCN treatment (*p* < 0.05). In addition, the *TNF-α* mRNA expression level was measured to detect the activation of the extrinsic pathway of apoptosis. GCN pretreatment markedly lowered the PM_2.5_-indued *TNF-α* mRNA expression level (*p* < 0.05) (Figure 8C). Furthermore, GCN pretreatment markedly reduced the protein expression levels of p-JNK, which can be phosphorylated by ROS or TNF- *α,* and p-c-Jun compared to the GRC- and PM_2.5-_treated A549 cells (*p* < 0.05) (Figure 8D). Collectively, these results suggest that GCN treatment effectively protects against apoptosis induced by PM_2.5_.

To identify the potential apoptotic pathways of target genes affected by PM, the analysis of the KEGG pathway was conducted using the Database for Annotation, Visualization, and Integrated Discovery (DAVID) [83]. Nine genes related to apoptosis and the mitogen-activated protein kinase (MAPK) signaling pathway were selected based on the DAVID KEGG analysis. PM_2.5_-induced apoptosis was shown to likely occur through the activation of mitochondria-mediated signaling pathways, which are triggered by ROS and TNF-α. According to our data and the KEGG pathway analysis, a schematic illustration of the potential signaling mechanisms of GCN attenuation against PM_2.5_-induced apoptosis was constructed (Figure 9).

## 3. Discussion

Previous studies have highlighted the health benefits of GRC, *R. nulubilis*, and *C. militaris*. These benefits include antioxidant, anti-cancer, immunomodulatory, and anti-inflammatory properties. A variety of active substances in GRC, such as cordycepin, adenosine, polysaccharides, flavonoids, and phenolic compounds, have been shown to effectively alleviate the harmful effects of PM_2.5_ on respiratory diseases [19,20,21,27,28,29].

The primary challenge of using GRC is its low bioavailability. Without protective processes such as encapsulation, the active compounds in GRC face significant drawbacks. For instance, cordycepin and adenosine, the key active components within *C. militaris*, can be converted into inactive metabolites due to their rapid metabolism in the body [95]. Similarly, phenolic compounds such as isoflavones and proanthocyanidins exhibit limited bioavailability because of their low water solubility, poor stability, and restricted membrane permeability [43,96,97]. To overcome these limitations, GRC has been encapsulated in CNPs to safeguard their components and enhance the delivery of hydrophilic and hydrophobic bioactive substances.

We synthesized optimal CNPs by applying ionotropic gelation with TPP as a crosslinker. In our previous study, we analyzed and established the ideal stoichiometric ratio between TPP and CHI to develop stable, narrow-sized CNPs for effectively encapsulating GRC [53].

Although encapsulation improves the protection and delivery of bioactive components of GRC, high concentrations of GRC extract can reduce the EE% due to competitive binding, diffusion, and weak adsorption. The decrease in the EE% with increasing GRC concentration is linked to higher levels of bioactive substances, including proteins, polysaccharides, and polyphenols, which reduce the available CHI for encapsulation [61,67,98]. Competitive binding occurs when proteins and polyphenols in GRC interact with CNPs, disrupting encapsulation. Proteins with net positive charges repel CHI but attract negatively charged TPP, whereas polyphenols destabilize the nanoparticle system (Figure 1B, competitive binding) [63,64]. High concentrations of the active compounds also increase the viscosity of the solution, slowing nanoparticle hardening, allowing diffusion losses, and further decreasing the EE% (Figure 1B, diffusion) [65]. In addition, the weak electrostatic adsorption of the GRC components on the CHI surface leads to detachment during centrifugation, reducing the EE% (Figure 1B, weak adsorption) [66,67]. Despite these challenges, the LE% increased with GRC concentrations from 2 to 8 mg/mL, with an optimal LE% of 7.62% achieved at 8 mg/mL [61,62].

The successful encapsulation of GRC in CNPs was confirmed using FT-IR spectroscopy and XRD analyses. FT-IR spectroscopy revealed ionic bonding between the amino groups of CHI and the polyphosphate groups of TPP [68,69], along with hydrogen bonding interactions with the GRC components [99,100]. These hydrogen bonds contribute to the stability and EE% of the nanoparticles. The disappearance of the amide I-II peaks in CHI along with the appearance of a modified peak at 1639 cm^−1^ in GCN indicate new electrostatic interactions between the -NH_3_^+^ groups of CHI and the -COO groups of GRC compounds, such as peptides from soybeans and *C. militaris* [101], and suggest tractable changes in the nature of the amine, amide, and carbonyl groups. Such interactions were further supported by quantitative analysis of the extent of carbonyl group conjugation. Mucsi et al. demonstrated that the carbonylicity and IR frequency were correlated, as an increase in carbonyl conjugation was associated with a shift in the carbonyl IR frequencies [102]. The amide I and II peaks of CHI appeared as a modified peak at 1639 cm⁻^1^ in GCN, highlighting the increase in carbonylicity due to the electrostatic interactions between CHI and GRC compounds. This increase in carbonylicity indicated an increase in the structural stability of nanoparticles. XRD analysis demonstrated a shift from a crystalline to an amorphous structure in CHI, supporting effective encapsulation [74].

Positively charged CNPs enhance cellular uptake by binding to negatively charged proteins on the cell surface, improving mobility, and facilitating membrane interactions [103,104]. This allows CNPs to act as intracellular drug reservoirs, releasing bioactive compounds in response to cellular acidity [103,104,105]. Enhanced absorption of FITC-GCN in the mouse small intestine in previous studies suggested improved bioavailability and efficacy of GRC [53]. The encapsulation of GRC in CNPs shows the potential for improving biological activities, including anti-inflammatory and antioxidant properties, alleviating lung injury and respiratory diseases.

This study investigated the effects of PM_2.5_ against lung damage, focusing specifically on oxidative stress and its subsequent influences on lung epithelial barrier integrity, inflammation, and fibrosis. Histopathological analysis indicated that PM_2.5_ exposure contributed to severe pulmonary damage, including increased inflammatory cell infiltration, alveolar wall thickening, hemorrhage, and airspace enlargement, which are characteristic of diseases such as emphysema, chronic bronchitis, asthma, and PF. These conditions are caused by alveolar wall destruction, inflammation, oxidative stress, apoptosis, and excessive proteolysis [106,107,108,109].

Previously, we reported that several bioactive compounds in GCN inhibit cellular apoptosis using their antioxidant effects [32]. Previous studies have reported that soy isoflavones (genistein and daidzein) in *R*. *nulubilis* (black soybean) inhibit H_2_O_2_-induced apoptosis by their ROS scavenging effect, repression of the apoptosis signaling pathway, and modulation of cell survival signaling [110,111]. Liu et al. reported that *Cordyceps* polysaccharides inhibit cellular apoptosis by their antioxidant properties and scavenging intracellular ROS [87]. GCN, which contains polysaccharides and isoflavones (daidzein, genistein, and glycitein), has been suggested to inhibit PM-induced apoptosis. This inhibition aids in maintaining alveolar unit homeostasis and prevents PM-induced damage to alveolar walls and epithelial cells [108,112]. Based on these findings, we confirmed that GCN pretreatment markedly attenuated severe lung injury with pathological variations induced by PM_2.5_. The following experiments were conducted to demonstrate the ameliorative effect of GCN against lung damage induced by PM_2.5_, with a particular focus on lung epithelial barrier damage.

Lung epithelial barrier integrity is maintained by the junctions between adjacent cells formed by TJ proteins [113]. Alterations in the function and composition of TJs play crucial roles in the pathogenesis of respiratory diseases [114]. TJ proteins include transmembrane proteins including zonula occludens, junctional adhesion molecules, claudins, E-cadherin, and occludin [115]. The occludin degradation induced by PM_2.5_ can result from various factors, including oxidative stress, mitochondrial membrane potentials (MMPs), and cell death [116,117]. Our findings indicated that GCN pretreatment restored the occludin protein levels, which were reduced by PM_2.5_, consistent with the protective effects of GCN on lung damage. Numerous studies have reported the protective effect of genistein on intestinal TJ barrier dysfunction against oxidative stress by activating protein tyrosine kinases and inhibiting protein tyrosine phosphatases, resulting in phosphorylated TJ proteins on tyrosine residues [118,119,120,121]. Thus, our results suggest that soy isoflavones in GCN, such as genistein, may attenuate occludin degradation by reducing oxidative stress through the inhibition of protein tyrosine kinase activation.

Similarly, the disrupted balance between proteolytic enzymes and their inhibitors is pivotal in alveolar destruction [122]. MMPs are proteolytic enzymes that cause significant changes in the lung extracellular matrix (ECM) by degrading its components, including collagen and elastin [123,124]. Lung tissue remodeling caused by MMPs is a characteristic of several lung diseases, including the alveolar wall destruction in emphysema and subepithelial fibrosis in the airways associated with asthma [124]. Several studies have reported that exposure to PM_2.5_ markedly upregulates MMPs, leading to air space expansion and alveolar wall destruction [125,126]. MMP-9 (gelatinase B) is typically present at low levels in healthy lungs but is markedly elevated in various lung diseases, including asthma, PF, and COPD [127]. PM-induced ROS inactivates anti-proteases and converts the proenzyme form of MMP-9 into its active form, leading to an imbalance with an excess of protease over anti-protease [107]. The resulting excessive proteolysis driven by oxidative stress is linked to emphysema [125,126]. In addition, PM stimulates inflammatory cells or resident pulmonary cells to secrete excess MMP-9, which degrades the ECM in the lungs and disrupts resident cells [128,129]. During PM-induced lung barrier destruction, we observed a corresponding decrease in *MMP-9* mRNA levels after GCN treatment. Several groups have reported that soy bioactive compounds in *R. nulubilis* such as isoflavones and saponins inhibit MMP activity [130,131]. In our previous study, *C. militaris* grown on germinated soybeans suppressed mRNA expression levels of *MMP-9* and *MMP-3* in colon cancer tissues [132]. Our results showed that GCN extract may help prevent alveolar wall destruction caused by PM_2.5_-induced excessive proteolysis.

Exposure to PM significantly increases the synthesis of collagen, a key component of the ECM [133]. The activation of MMP-9 and collagen accumulation in PM-exposed lung tissues suggested that PM exposure triggered the activation of pulmonary fibroblasts, leading to ECM accumulation [134]. Abnormal and excessive accumulation of ECM components, such as *CoL1A1*, is a key characteristic of PF, resulting in tissue destruction and impaired lung function [135]. The increased *MMP-9* and *COL1A1* mRNA expression levels correlated with the alveolar wall thickening observed in this study, indicating an abnormal healing process following lung tissue destruction and dysfunction due to PM exposure [136]. Although the molecular mechanisms underlying PM-induced PF are not yet fully understood, previous studies have identified that stimulation of TGF-β signaling and fibroblasts, leading to excessive collagen deposition in lung tissue, are characteristic features of PF [137,138,139]. Future studies will focus on the impact of GCN on activated fibroblasts and excessive TGF-β, which are essential for collagen synthesis and contribute to the development of PM-induced PF [133]. This indicates that the isoflavones, including daidzein and genistein, present in GCN may play a role in its antifibrotic activity by reducing collagen deposition in the lungs [140,141,142].

PM_2.5_, or fine particulate matter, is a significant environmental pollutant associated with various lung diseases, primarily because of its ability to trigger oxidative stress and inflammation [143]. The smaller particle size of PM_2.5_ allows it to penetrate deep into the lungs, reaching the alveoli, where it can interact with lung cells and exacerbate respiratory conditions [9,144,145,146]. PM_2.5_ contains higher concentrations of heavy metals than larger particles, causing elevated ROS production and oxidative DNA damage. These interactions can result in a range of harmful health effects, including the induction of apoptosis, which contributes to lung tissue destruction and dysfunction [147,148,149].

The chemical composition of PM_2.5_ includes organic compounds like PAHs, inorganic components like metals and ions, and biological materials [150]. In particular, PAHs induce activation of the aryl hydrocarbon receptor that upregulates cytochrome P450 enzymes, which causes the generation of highly toxic metabolites and ROS [151,152]. Transition metals in PM_2.5_, such as iron and copper, can undergo Fenton and Haber–Weiss reactions to catalyze the highly reactive hydroxyl radical formation, further promoting oxidative stress [12]. These processes result in cellular damage, DNA strand breaks, and destruction of alveolar walls, which are characteristic of conditions such as emphysema [143,153,154,155].

To counteract oxidative stress induced by PM_2.5_, antioxidants are crucial in mitigating cellular damage. Natural compounds, such as genistein from *R. nulubilis* and polysaccharides from *C. militaris*, have been shown to possess antioxidant properties. Genistein scavenges hydrogen peroxide and inhibits the formation of superoxide anions [93], whereas polysaccharides from *C. militaris* exhibit radical scavenging abilities and metal-chelating activities, particularly with Fe^2^⁺ ions [88,89,156]. These antioxidants help reduce ROS levels and defend lung cells against oxidative stress.

In addition to direct ROS production, PM_2.5_ exposure also triggers inflammatory responses in the lungs. Alveolar macrophages, which are key immune cells in the lungs, release pro-inflammatory cytokines such as TNF-α in response to PM_2.5_ phagocytosis [157]. This causes inflammation and oxidative stress, inducing tissue dysfunction and cell death.

PM_2.5_ exposure also impairs the lung antioxidant defense system by downregulating the Nrf2/ARE pathway, which regulates the expression of antioxidant enzymes including catalase and superoxide dismutase (SOD) [158,159,160,161]. Nitric oxide (NO), synthesized by iNOS in response to PM exposure, combines with superoxide, forming peroxynitrite, a potent oxidant that contributes to lung tissue damage [162,163,164]. This dysregulation of antioxidant defenses makes the lung tissue more vulnerable to oxidative stress and further exacerbates conditions such as COPD [165,166].

Cordycepin and adenosine in *C. militaris* respectively increase the levels and activities of antioxidant enzymes including glutathione peroxidase and SOD [84,85]. Furthermore, it has been reported to suppress iNOS expression, thereby inhibiting NO production, owing to its anti-inflammatory properties [20,86]. It has also been reported that soy isoflavones in *R. nulubilis* have NO and ONOO scavenging effects [167]. We observed that GCN significantly improved *SOD1* and *iNOS* mRNA expression levels in murine lungs compared with GRC, suggesting that isoflavones and cordycepin in GCN enhanced antioxidant levels in the lungs, whereas the hydrophilic CNPs used for GRC encapsulation improved the antioxidant activities of water-soluble compounds, such as isoflavones, in GRC by facilitating higher loading and enhancing cellular absorption [168].

Numerous studies have explored the protective effects of genistein and polysaccharides from *C. militaris* to counteract the harmful effects of PM_2.5_ by reducing lung cell apoptosis induced by PM_2.5_ [87,94,111]. These compounds have shown promise in reducing intracellular ROS levels and enhancing antioxidant defenses, suggesting that they may offer therapeutic potential in preventing or mitigating lung damage caused by PM_2.5_ exposure [87,110,111].

Furthermore, the study delves into the mechanisms of apoptosis induced by PM_2.5_, highlighting the critical roles of inflammation and oxidative stress in triggering both intrinsic and extrinsic apoptotic pathways [11,169,170,171].

PM-induced oxidative stress initiates intrinsic apoptosis in epithelial cells and lung tissues by disrupting mitochondrial function [169]. Mitochondrial dysfunction triggered by PM causes a considerable reduction in the MMP, which subsequently alters the Bax and Bcl-2 protein expression levels. This activates the caspase cascade and PARP cleavage, culminating in apoptosis [170,172]. c-Jun NH2-terminal kinase (JNK) pathway activation is induced by oxidative stress, leading to intrinsic apoptosis. In PM_2.5_-exposed lung epithelial cells, ROS generation activates apoptosis signaling kinase 1 (ASK1), stimulating the JNK pathway and leading to p53 phosphorylation and activation [173,174]. Activated p53 regulates the genes associated with apoptosis, DNA repair, and cell cycle regulation. Upon activation of p53, the upregulation of proapoptotic genes (such as Bax, Bak, and Bid) and the downregulation of antiapoptotic genes (such as Bcl-2 and Bcl-XL) are induced [175]. Proapoptotic proteins in the cytosol translocate to the mitochondrial membrane, inducing cytochrome c (cyt c) release into the cytosol [11]. This release forms the apoptosome by binding cyt c to apoptotic protease-activating factor-1 (Apaf-1) and procaspase-9, activating caspase-9, which then initiates the caspase-3 cascade and culminates in apoptosis [176]. Apoptotic cell death and imbalance in cellular homeostasis can lead to alveolar septal destruction [108,112].

In addition to oxidative stress, inflammation also contributes to alveolar septal destruction by increasing apoptosis rates [108]. PM exposure elevates inflammatory cell infiltration and pro-inflammatory cytokines, notably, IL-6 and TNF-α, in lung epithelial cells, triggering TNF-α-induced apoptosis. TNF-α binds to TNFRI and TNFRII receptors, activating caspase-8 and, subsequently, caspase-3, driving extrinsic apoptosis [11,170,171,177,178,179].

*Cordyceps* polysaccharides, a major antioxidant component of *C. militaris*, effectively scavenge ROS and regulate Cyt c, Bax, and Bcl-2 apoptotic proteins to protect hepatocytes from apoptosis caused by H_2_O_2_-induced mitochondrial dysfunction [87]. Adenosine in *C. militaris* suppresses ROS generation, regulates Bcl-2/Bax levels, and decreases MAPK phosphorylation [85]. Soy isoflavones from *R. nulubilis* inhibit ROS production and regulate the mitochondria-mediated apoptotic pathway. Genistein inhibits H_2_O_2_-induced apoptosis by attenuating JNK phosphorylation [94,111]. In our study, GCN pretreatment suppressed apoptotic signaling molecules in A549 cells, suggesting that the isoflavones, adenosine, and polysaccharides in GCN reduce PM_2.5_-induced intrinsic apoptosis by scavenging excessive ROS and regulating apoptotic signals. Furthermore, the anthocyanins in R. *nulubilis*, especially cyanidin-3-O-glucoside, block various inflammatory signaling pathways by inhibiting the phosphorylation of JNK and p38 and the levels of NF-κB and suppressing the expression of pro-inflammatory cytokines like IL-6, IL1β, and TNF-α, potentially helping to suppress TNF-α-induced extrinsic apoptosis [90,91,92].

Although *C. militaris* shows anti-inflammatory and antioxidant activities, its narrow TI and cytotoxicity limit its efficacy [180]. Natural compounds often have low TIs, where the effective dose closely approaches the toxic dose, and many are hydrophilic, leading to low cellular uptake, enzyme degradation, rapid clearance, and limited efficacy [181,182,183]. The hydrophilic compounds of GRC, such as isoflavones, anthocyanins, and polysaccharides, demonstrate antiapoptotic effects but require enhanced bioavailability to improve their safety–efficacy profile [81]. This study found that encapsulating GRC in chitosan nanoparticles (GCN) improved the TI, enhancing antiapoptotic effects and reducing A549 cell death. The cationic nature of CHI allows for electrostatic interactions with negatively charged cell membranes, enhancing the cellular uptake of hydrophilic GCN compounds, and thus enhancing antioxidant efficacy [54]. CNPs also protect the encapsulated GRC molecules from degradation, further improving their bioavailability [103]. These results suggest that GCN enhances bioavailability and induces antiapoptotic effects to prevent PM_2.5_-induced lung injury.

To conclude, this study emphasized the complex interplay between apoptosis, inflammation, and oxidative stress underlying lung damage triggered by PM_2.5_. It also highlights the potential of antioxidant compounds, such as those found in *R. nulubilis* and *C. militaris*, to mitigate PM_2.5_-induced harmful effects and protect lung tissue from oxidative damage and cell death.

## 4. Materials and Methods

### 4.1. Preparation of GCN

GCN was synthesized by employing the ionic crosslinking technique using TPP, as stated before [53]. For the preparation of 4 mg/mL CHI solution, CHI powder (200 mg) was dissolved with deionized water (50 mL). CHI was fully dissolved by introducing 0.5 mL of acetic acid in a dropwise manner over 1 h, with magnetic stirring. After homogenization, the CHI solutions were neutralized to pH 5.5 using sodium hydroxide (0.5 M), followed by filtration through a 0.45 µm filter. Sodium triphosphate was solubilized in deionized water to 1 mg/mL and was neutralized to pH 2 using hydrochloric acid (1 M). The solution was subjected to filtration through a 0.45 µm filter. GCN was synthesized by adding GRC extract (8 mg/mL), which was prepared by patented technology (Cell Activation Research Institution, Seoul, Republic of Korea; voucher specimen Kucari: 0903), under optimal conditions (CHI:TPP (4:1 *w*/*w*), TPP pH 2). Specifically, 6 mL of the CHI solution (4 mg/mL) was mixed with 6 mL of various concentrations (2, 4, 8 mg/mL) of GRC and stirred at 600 rpm for 5 min. Using a NE-300 syringe pump (New Era Pump Systems Inc., Farmingdale, NY, USA), 6 mL of the TPP solution (pH 2) was gradually introduced to the CHI solution under stirring at a controlled rate of 1 mL/min, a process lasting 6 min. Following an additional 30 min of stirring, the solution underwent centrifugation in a high-speed centrifuge (model Avanti J-E; Beckman Coulter, Brea, CA, USA) at 15,000 rpm for 30 min. Following centrifugation, the pellet was resuspended with 5 mL of a 10% sucrose solution prior to freeze-drying, and the supernatant was reserved to measure the unencapsulated GRC content.

### 4.2. EE% and LE%

The EE% and LE% of GCN were calculated indirectly. Briefly, the amount of unencapsulated GRC was determined as reported by Papadimitriou et al., as well as by Manne et al. [68,69]. After centrifuging the GCN solution, the amount of unencapsulated GRC in the supernatant was evaluated based on measuring the absorbance (OD) at 380 nm (λ max) with a calibration curve (8 mg/mL GRC concentration). The OD value was assessed using an Epoch microplate reader (BioTek Instruments, Inc., Winooski, VT, USA). The amount of GRC extract entrapped in nanoparticles was calculated using Equation (1):Amount of GRC extract entrapped in nanoparticles = Initial amount of GRC extract-amount of unencapsulated GRC extract calculated from the supernatant(1)

EE% and LE% of GCN were determined using Equations (2) and (3), respectively:(2)$$Entrapment\;efficiency\left(\%\right)=\frac{amount\;of\;GRC\;extract\;entrapped\;in\;nanoparticles}{theoretical\;amount\;of\;GRC\;extract\;entrapped\;in\;nanoparticles}\times 100$$(3)$$Loading\;of\;efficiency\left(\%\right)=\frac{amount\;of\;GRC\;extract\;entrapped\;in\;nanoparticles}{weight\;of\;nanoparticles}\times 100$$

### 4.3. In Vitro Release

The in vitro release profile was determined by suspending freeze-dried GCN in 0.5 mL of buffer solution (CHI:TPP mass ratio = 4:1, TPP at pH 2, containing 2.5 mg of GRC), placing it into a dialysis bag, and immersing it in 50 mL of PBS, with a pH of 2.2, 5.5, or 7.4, contained in a beaker. These pH values were selected to simulate those of gastric fluid (pH 2.2), skin/storage conditions (pH 5.5), and intestinal/blood conditions (pH 7.4). This solution was incubated at 100 rpm in a shaker bath (BS-11; Shaking Water bath; Jeio Tech Co., Ltd., Daejeon, Republic of Korea) at 37 °C. To evaluate drug release over time, 0.5 mL aliquots were collected at each time point (0, 0. 5, 2, 4, 24, 48, and 72 h), and the aqueous suspension was separated by centrifugation at 10,000 rpm for 5 min at 4 °C and suspended in buffer (500 μL). To maintain sink conditions, any removed sample solution was replenished with fresh PBS. This procedure prevents the accumulation of released GRC and ensures sink conditions, as recommended for reliable in vitro drug-release studies [184]. The amount of GRC released was measured using an ultraviolet–visible (UV-vis) spectrophotometer at a wavelength of 380 nm. The same volume of fresh buffer was incorporated into each mixture. The cumulative percentage of GRC released from GCN was evaluated using Equation (4):(4)$$Cumulative\;release\left(\%\right)=\frac{cumulative\;amount\;of\;released\;GRC\;at\;each\;time\;interval}{initial\;weight\;of\;the\;GRC\;loaded\;in\;the\;sample}\times 100$$

### 4.4. Synthesis of FITC-Labeled GRC and GCN

FITC-labeled GRC extract and GCN were synthesized as previously described [185]. GRC (8 mg/mL) and CHI (4 mg/mL) were used as outlined in Section 4.1 (Preparation of GCN). The synthesis was conducted using a CHI:TPP of 4:1 *w*/*w*, with TPP solution (1 mg/mL, pH 2). By dissolving FITC powder in 5 mL of DMSO (0.25 mg/mL), a final FITC concentration of 0.05 mg/mL was achieved. To prepare the FITC-labeled GRC extract, 200 µL of FITC solution was combined with 300 µL of GRC (8 mg/mL). FITC-GCN was synthesized by incorporating 300 μL of the FITC-labeled GRC extract into the CHI solution (4 mg/mL) and subsequently adding 300 μL of TPP solution. The mixture was uniformly stirred using a pipette. After centrifuging (15,000 rpm, 30 min) FITC-GCN and FITC-GRC, the resuspension of the pellets in 1 mL of deionized water was carried out. This step was repeated until the supernatant no longer showed any detectable fluorescence.

### 4.5. PM Sample and Cell Culture Preparation

The technique for collecting and extracting PM_2.5_ was adapted from protocols detailed in an earlier study [32]. For the preparation, HEPA filters were used to collect the fine particles. Briefly, the filters were immersed in a 50 mL tube containing 75% ethanol (10 mL) and subjected to sonication (30 min, 4 °C) using an Ultrasonic Processor sonicator (SONICS, Newtown, CT, USA). Subsequently, particles smaller than 2.5 μm were isolated using a Whatman filter paper (1005-055, Maidstone, Kent, UK). The resulting suspension was then concentrated under reduced pressure and stored at −80 °C for further experimentation (Figure 10). A549 cells were purchased from the Korean Cell Line Bank (KCLB, Seoul, Republic of Korea). Cells were maintained in Roswell Park Memorial Institute (RPMI) 1640 medium (Welgene, Seoul, Republic of Korea) with 10% fetal bovine serum (FBS) and 1% penicillin–streptomycin (Welgene). The cells were cultured in 75 cm^2^ flasks at 37.5 °C in a humidified atmosphere containing 5% CO_2_.

### 4.6. In Vivo PM Exposure

All animal procedures were approved by the Institutional Animal Care and Use Committee (IACUC) of Gachon University (approval number: GU1-2022-IA0023-01) in compliance with relevant guidelines. Six-week-old female BALB/c mice were obtained from JA BIO (Suwon, Gyeonggi, Republic of Korea) and acclimated under controlled conditions of 20 ± 2 °C and a 12 h light/dark cycle (lights on at 06:00), with free access to food and water. Post-acclimatization, the mice were divided into four groups (*n* = 6), following a protocol inspired by Vignal et al. Using a PARI BOY^®^ SX nebulizer (PARI GmbH, Starnberg, Germany), a PM solution with a concentration of 132.8 µg/mL was aerosolized into an 11 L chamber over four weeks (5 days/week). This setup delivered 796.8 µg of PM daily at a concentration of 1.66 µg/L, while control animals were exposed to distilled water [186]. An additional two-week period involved intranasal exposure to PM, with 8 µg delivered in 40 µL of PBS per application (Figure 11).

Seoul’s PM level averages 27 µg/m^3^, which equates to a daily human exposure dose of 235.87 µg (Table 1) [187,188]. Considering respiratory volumes, mice require exposure to around 8 μg of PM per day to simulate Seoul’s air pollution levels [188]. The mice were administered 100 μL PBS or 300 mg/kg of GCN and GRC samples via oral gavage for 6 weeks (5 days/week). The animals were subjected to anesthesia through an intraperitoneal injection of Avertin (100 mg/kg) followed by euthanasia via cardiac puncture at the final stage. Lung tissues, after harvesting, were preserved at −80 °C for molecular assays (western blot and RT-PCR), and for histological analysis, they were fixed in 10% formaldehyde and embedded in paraffin.

### 4.7. Evaluation of Lung Injury Through H&E Staining

Deparaffinization of lung tissue sections on slides was performed, followed by H&E staining, according to the protocol provided by the manufacturer (TissuePro Technology, Gainesville, FL, USA). Following staining, the tissue slides were observed using a Nikon Eclipse Ti-S microscope (100×). Lung damage severity was determined using the lung injury score based on the criteria established by Matute-Bello’s group [189,190]. As shown in Table 2, lung sections were separately evaluated on a 0–3 scale across four criteria: intra-alveolar infiltrations, intra-alveolar fibrin, alveolar hemorrhage, and alveolar septa. For the lung injury score, five different fields for each mouse and three mice from each group were evaluated by three blinded investigators. Representative values for each group were obtained by calculating the mean values.

### 4.8. Acute Toxicological Study

The acute toxicity of GRC and GCN was carried out to evaluate the safety in accordance with the guidelines of the National Institute of Food and Drug Safety Evaluation (NIFDS) [191]. The female BALB/c mice were randomly allocated into three groups (*n* = 5 per group). Over a two-week period, GCN or GRC (2000 mg/kg) was administered to each group via oral gavage. The same volume of PBS was administered to the control group. The mice were monitored for two weeks to observe the general symptoms, record the body weight changes, and evaluate the visual inspection. Body weight was measured daily using a tabletop electronic balance.

### 4.9. Cell Viability Assay

A CCK-8 kit (Dojindo Laboratories, Kumamoto, Japan) was employed to evaluate A549 cell viability, following previously established protocols [32,192]. Cells were plated in 96-well plates at 1 × 10^4^ cells per well. To compare the effects of GCN and GRC on A549 cell viability, cells were treated with 50, 100, or 200 µg/mL of GCN or GRC for 1 h, followed by treatment with 200 µg/mL of PM_2.5_. Following a 24 h incubation, CCK-8 solution (10 µL) was dispensed into each well and incubated with A549 cells for 1 h. The absorbance at 450 nm was quantified using a microplate reader. The *ED50* and *LD50* were analyzed using ED50plus v1.0 software, which employs straight-line regression for cumulative concentration–response curves expressed as μg/mL. The software was designed by Mario H. Vargas from the National Institute of Respiratory Diseases in Mexico. In addition, the therapeutic index (*TI*) was calculated using Equation (5):(5)$$TI=\frac{LD50}{ED50}$$

### 4.10. Immunohistochemistry

An immunohistochemical analysis was performed according to methods outlined in prior studies [193]. This involved using the primary antibody SOD1 at a 1:200 dilution (provided by Cell Signaling Technology Inc., Danvers, MA, USA) and incubating for 1 h at RT. After the preceding step, the tissue samples were washed and treated for 30 min with an anti-rabbit secondary antibody conjugated to biotin, sourced from DAKO, Carpinteria, CA, USA. Detection of the secondary antibody was achieved by applying a streptavidin–HRP conjugate for 30 min. The detection of antibodies was achieved using a DAB chromogen kit provided by Vector Laboratories (Burlingame, CA, USA), and counterstaining was performed with methyl green (1%, 1 min). Imaging was conducted with a Nikon Eclipse Ti microscope integrated with a Point Grey Research digital camera (Richmond, BC, Canada). Image analysis was conducted using MetaMorph software NX 2.0 by Molecular Devices (Sunnyvale, CA, USA).

### 4.11. Reactive Oxygen Species (ROS) Assay

ROS levels were evaluated according to a previously reported protocol [32]. DCFH-DA, a reactive probe that interacts with ROS to yield the fluorescent compound 2,7-dichlorofluorescein (DCF), was used to evaluate intracellular ROS levels, following the manufacturer’s instructions (ab113851, Abcam, Cambridge, UK). A549 cells were plated in 12-well plates at 2 × 10^5^ cells per well and incubated overnight. Exposure to GCN and GRC was conducted for 1 h before exposure to 200 μg/mL of PM_2.5_ for 3 h. The cells were exposed to DCFH-DA (25 μM) and incubated for 45 min at 37 °C, allowing DCFH-DA to react with intracellular ROS. The fluorescence emitted by the stained cells was captured using a Nikon Eclipse Ti microscope at 100× magnification, using MetaMorph software NX 2.0 by Molecular Devices (Sunnyvale, CA, USA) for image acquisition. Quantification of fluorescence intensity, indicative of ROS levels within the cells, was performed using ImageJ software v1.54k. Hydroxyl peroxide served as a positive control to stimulate ROS generation. The results are presented as fold changes compared to the control.

### 4.12. RNA Isolation and RT-PCR

RNA extraction was performed on tissue samples and A549 cells exposed to PM_2.5_, using a method described in prior studies [192]. The PCR reaction involved an initial denaturation at 94 °C for 2 min, followed by 30 cycles of denaturation (94 °C, 30 s), annealing (55 °C, 30 s), and extension (68 °C, 1 min). Primers spanning a range of sequences were employed to amplify specific gene targets: mouse *COL1A1* (forward: 5′ TGG TCC ACA AGG TTT CCA AG 3′, reverse: 5′ TTC ACC CTT AGC ACC AAC TG 3′), mouse *MMP-9* (forward: 5′ AGG CCT CTA CAG AGT CTT TG 3′, reverse: 5′ CAG TCC AAC AAG AAA GGA CG 3′), mouse *SOD-1* (forward: 5′ CAT CCA CTT CGA GCA GAA GG 3′, reverse: 5′ CAA TCA CTC CAC AGG CCA AG 3′), mouse *iNOS* (forward: 5′ CTT CAA CAC CAA GGT TGT CTG CA 3′, reverse: 5′ ATG TCA TGA GCA AAG GCG CAG AA 3′), and mouse *GAPDH* (forward: 5′ GAA GGT CGG TGT GAA CGG AT 3′, reverse: 5′ ACT GTG CCG TTG AAT TTG CC 3′). To ensure reliable quantification of gene expression changes, the levels of target genes were normalized against the housekeeping gene *GAPDH*.

### 4.13. Western Blot Analysis

Western blot analysis was performed following protocols previously described in the studies [53,194]. Using RIPA buffer (Cell Signaling Technology, Danvers, MA, USA), tissues and cells were lysed and homogenized to extract proteins. Centrifugation at 14,000× *g* for 10 min was employed to isolate proteins from the lysates. A Pierce BCA protein assay kit (Thermo Fisher Scientific, Waltham, MA, USA) was used to determine protein concentrations, and 10% sodium dodecyl sulfate–polyacrylamide gel electrophoresis (SDS-PAGE) was used to separate equal amounts of proteins. After separation, the proteins were transferred to nitrocellulose membranes (Bio-Rad Laboratories, Inc., Hercules, CA, USA) and incubated with 5% BSA for 1 h at RT. The membranes were incubated overnight at 4 °C in Tris-buffered saline with Tween buffer (TBS-T, 20 mM Tris, 500 mM NaCl, pH 7.6, 0.1% Tween 20) containing primary antibodies against Bax, Bcl-2, p53, caspase-9/-8/-3, and Occludin, all diluted to 1:1000 as per manufacturer protocols (Cell Signaling, Danvers, MA, USA). The blots were visualized using an enhanced chemiluminescence detection solution (EzWestLumi plus, ATTO corporation, Tokyo, Japan), and imaging was performed using Odyssey LCI Image software version 4.0 (LI-COR Biosciences, Lincoln, NE, USA). The blots shown are representative of at least 3 repeats.

### 4.14. Statistical Analysis

Data were derived from at least three independent replicates and are reported as mean ± SD. Multiple comparisons were conducted via one-way analysis of variance (ANOVA) followed by Tukey’s Honestly Significant Difference (HSD) test with a significant level of *α*  =  0.05 and a 95% confidence interval and an independent *t*-test with a significant level of *α*  =  0.01 and 99% as a confidence interval. Statistical data analysis was conducted using SPSS software, version 12 (IBM Corp., Chicago, IL, USA).

## 5. Conclusions

In summary, our study demonstrated the protective effects of GCN on lung dysfunction induced by PM_2.5_ in vivo and in vitro. PM_2.5_ exposure compromises lung barrier integrity and causes inflammation and oxidative stress, which exacerbate lung injury. GCN contains various antioxidant, anti-inflammatory, and antiapoptotic bioactive compounds, including cordycepin, polysaccharides, isoflavones, and anthocyanins. Therefore, GCN reduced PM_2.5_-induced oxidative stress in murine lungs and A549 cells by increasing antioxidant enzyme expression and intracellular ROS scavenging activity. In addition, GCN decreased inflammation by downregulating *iNOS* and *TNF-α* mRNA expression levels. Consequently, GCN mitigated ROS and inflammation-driven apoptosis in PM_2.5_-treated murine lungs and A549 cells by inhibiting JNK phosphorylation, thereby suppressing the expression of apoptotic signaling proteins. The encapsulation of GRC in chitosan nanoparticles protects its bioactive compounds from degradation, enhances mucoadhesion and cellular uptake, and improves its bioavailability and safety–efficacy profile. These findings indicate that GCN is a promising therapeutic candidate for the management of PM_2.5_-induced lung injury and respiratory diseases.

## Figures and Tables

**Figure 2 ijms-26-01105-f002:**
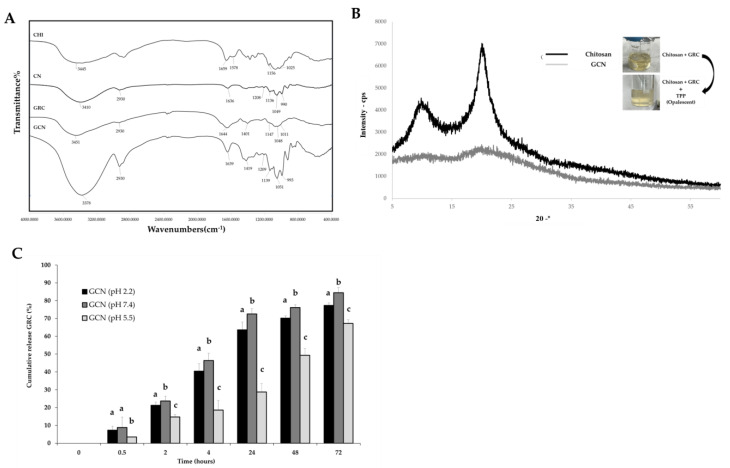
GRC chitosan nanoparticle (GCN) characterization. (**A**) FT-IR of CHI, chitosan nanoparticle (CNP), GRC, and GCN, (**B**) XRD graph of CHI and GCN, and (**C**) in vitro release profiles of GRC from GCN in different pH solutions (pH 2.2, 5.5, and 7.4) at 37 °C. Values are means ± standard deviations (SDs) of triplicate determinations. Lower-case letters indicate significant differences (Tukey HSD test *p* < 0.05).

**Figure 3 ijms-26-01105-f003:**
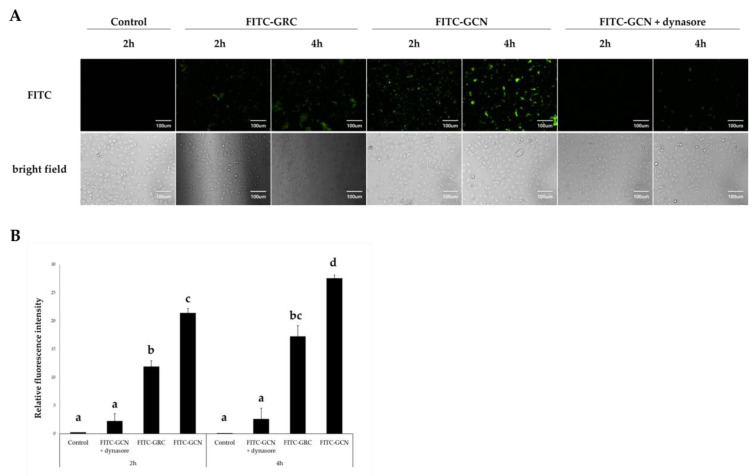
GRC encapsulated in CNPs (GCN) internalization of A549 cells. We treated 200 μg/mL of FITC-GRC or FITC-GCN and observed them using a fluorescence microscope after 2 or 4 h. We treated 200 μg/mL of GCN under the same conditions after inhibiting clathrin-mediated endocytosis of GCN by treatment with 80 μM dynasore for 30 min. Each group was observed with fluorescence microscopy after being washed twice with phosphate-buffered saline (PBS). (**A**) A549 cells were observed using confocal microscopy using Metamorph software NX 2.0 (Universal Imaging, West Chester, PA, USA; magnification = 200×; scale bar = 100 µm). (**B**) Relative fluorescence intensities of dynasore + FITC-GCN, FITC-GRC, and FITC-GCN were analyzed using ImageJ software v1.54k. Values are means ± SDs of triplicate determinations. Lower-case letters indicate significant differences (Tukey HSD test *p* < 0.05).

**Figure 4 ijms-26-01105-f004:**
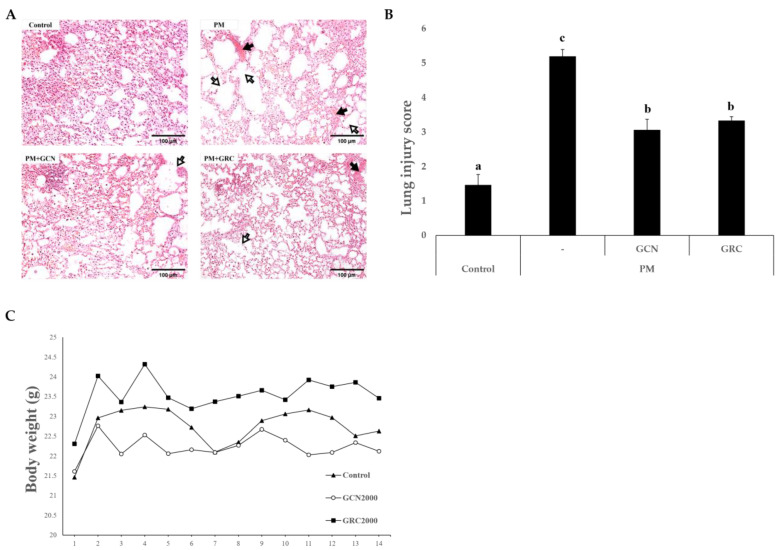
Mouse body weight evaluation during the acute toxicology test for the safety assessment of GCN and histological analysis of lung damage triggered by PM_2.5_. exposure (**A**) Hematoxylin and eosin staining of lung tissues (magnification = 100×; scale bar = 100 µm). The arrow represents capillary congestion, and the hollow arrow represents collapsed alveolar space. (**B**) Lung injury score in the indicated groups based on H&E staining. (**C**) Mouse body weight in the control, GCN, and GRC groups. The mice were orally treated with 2 g/kg of GCN and 2 g/kg of GRC. Each symbol corresponds to a PBS, GCN, or GRC formulation administered to the mice. Values are means ± standard errors (SEs). Lower-case letters indicate significant differences (Tukey HSD test *p* < 0.05).

**Figure 5 ijms-26-01105-f005:**
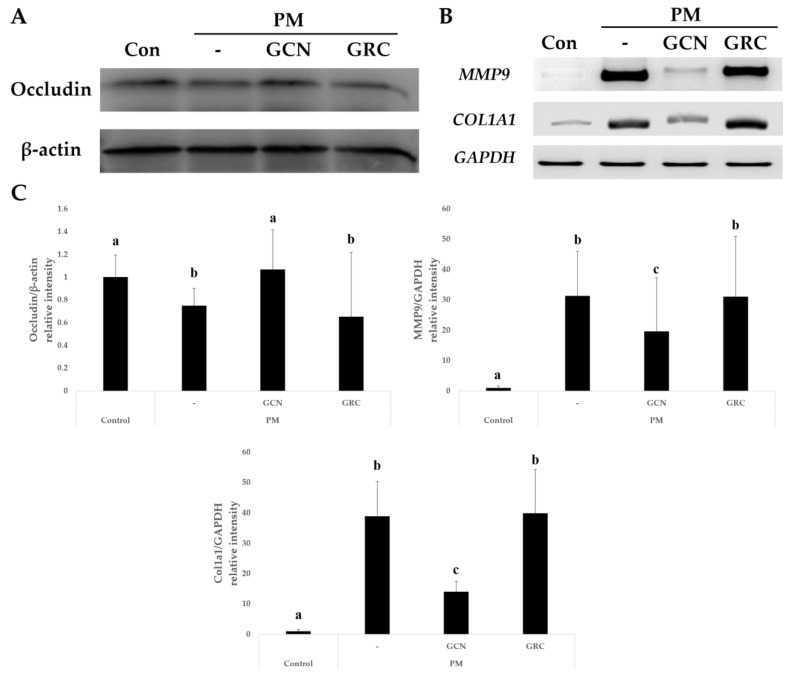
GCN reduces PM_2.5_-induced tight junction protein degradation and overexpression of *MMP9* and *COL1A1* mRNA in the murine lung. (**A**) Effect of GCN on the occludin protein expression level. (**B**) Effect of GCN on the levels of *MMP-9* and *COL1A1* mRNA expression. (**C**) Relative intensities of protein expression of occludin and mRNA expressions of *MMP9* and *COL1A1*. Values are means ± SEs of triplicate determinations. Lower-case letters indicate significant differences (Tukey HSD test *p* < 0.05).

**Figure 6 ijms-26-01105-f006:**
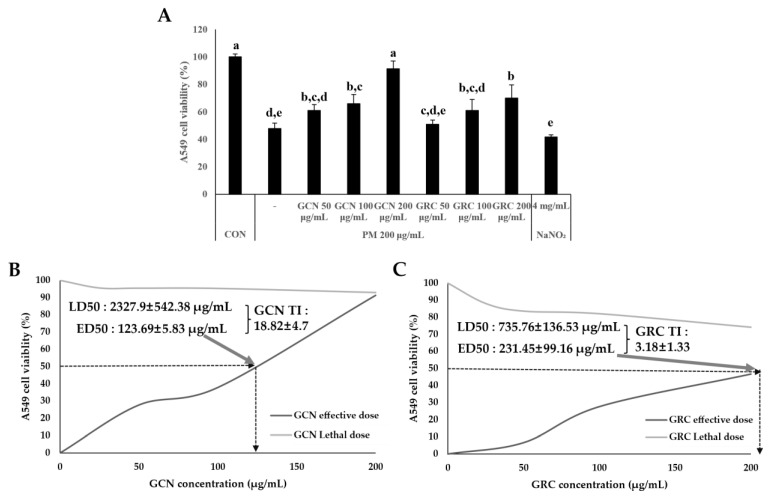
Effect of GCN on A549 cell viability. (**A**) Effect of GCN and GRC on PM_2.5_-exposed A549 cell viability (200 µg/mL). A549 cells were treated with different concentrations (50, 100, and 200 μg/mL) of GCN and GRC for 24 h. Sodium nitrite (NaNO_2_) was used as a positive control. A549 cell viability was assessed using a CCK-8 assay. (**B**,**C**) Dose–response curve of GCN and GRC on A549 cell viability. Values are means ± SDs. Lower-case letters indicate significant differences (Tukey HSD test *p* < 0.05). An independent sample *t*-test was performed to compare the data between two groups (*p* < 0.01).

**Figure 7 ijms-26-01105-f007:**
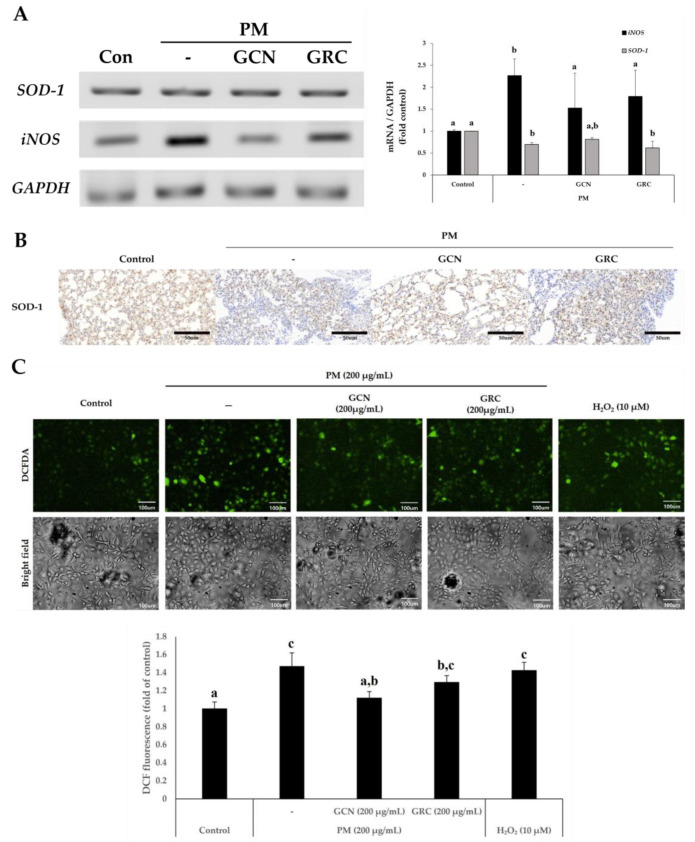
Antioxidant effects of GCN in particulate matter (PM_2.5_)-treated mice and A549 cells exposed to PM_2.5_ (200 µg/mL). (**A**) GCN-induced changes in oxidative stress-related mRNA expression (*SOD-1* and *iNOS*) in mouse lung tissue. (**B**) Expression of antioxidant enzymes in PM_2.5_-exposed mice lung tissue, accompanied by representative immunohistochemical staining images for SOD-1 (magnification = 200×, scale bar = 50 µm). (**C**) Intracellular ROS were assessed using fluorescence microscopy (Nikon Eclipse Ti microscope, Point Grey Research, Richmond, BC, Canada) after 2-7-dichlorodihydrofluorescein diacetate (DCF-DA) staining. Hydrogen peroxide (H_2_O_2_), a ROS inducer, was used as a positive control. PM_2.5_-induced ROS scavenging effect of 200 µg/mL of GCN and 200 µg/mL of GRC in A549 cells. The ROS degeneration ability was measured using DCF-DA, as described in the Materials and Methods Section, at a wavelength of 485/535 (Ex/Em). Values are means ± SDs of triplicate determinations. Lower-case letters indicate significant differences (Tukey HSD test *p* < 0.05). Control = PBS-treated mice, negative control group = particulate matter and PBS-treated mice, GCN = particulate matter and GCN (300 mg/kg/day)-treated mice, and GRC = particulate matter and GRC (300 mg/kg/day)-treated mice.

**Figure 8 ijms-26-01105-f008:**
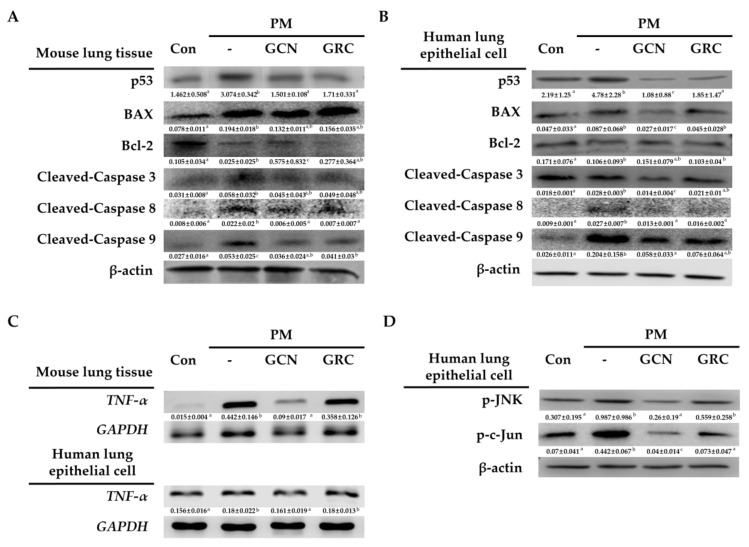
GCN inhibits the activation of PM_2.5_-induced apoptotic pathways. (**A**,**B**) BAX, BCL-2, p53, active caspase-3, active caspase-9, and active caspase-8 protein expression levels in lungs and cells were measured using WB. (**C**) *TNF-α* mRNA expression in both lungs and cells was determined using RT-PCR. (**D**) WB analysis of phosphorylated JNK and c-Jun in cells. Values are means ± SDs of triplicate determinations. Lower-case letters indicate significant differences (Tukey HSD test *p* < 0.05).

**Figure 9 ijms-26-01105-f009:**
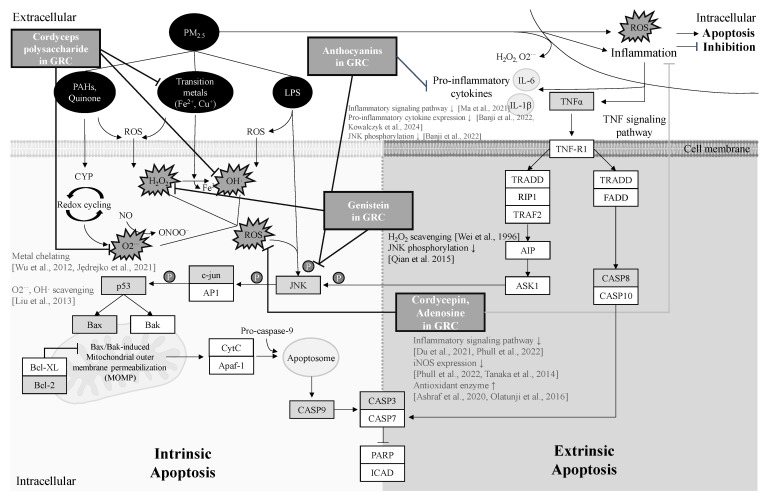
Summary of the apoptosis signaling pathway triggered by PM_2.5_-induced oxidative stress and inflammation. White boxes show TNF signaling and MAPK signaling pathway-related genes identified through KEGG pathway analysis using DAVID online tools (v2023q4); light gray boxes show nine identified genes associated with the TNF signaling pathway and the MAPK signaling pathway selected by KEGG pathway analysis. The Bold blunt arrows show the suppressive effects of cordycepin, adenosine [19,20,84,85,86], polysaccharide [87,88,89], anthocyanins [90,91,92], and genistein [93,94] in GRC against PM_2.5_-induced apoptotic pathway by mitigating ROS and inflammation.

**Figure 10 ijms-26-01105-f010:**
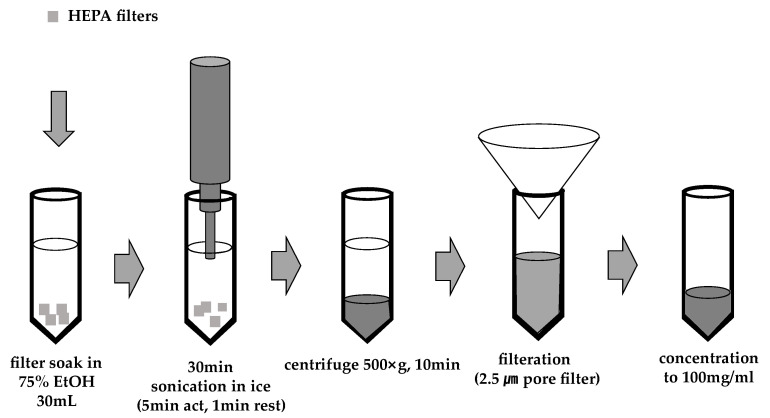
Process diagram of PM_2.5_ extraction. PM_2.5_ particles were collected from the campus of Gachon University, located in Seongnam-si, Gyeonggi-do, Republic of Korea. The collected samples were passed through a filter paper with a 2.5 µm pore size (1005-055, Whatman, Maidstone, Kent, UK).

**Figure 11 ijms-26-01105-f011:**
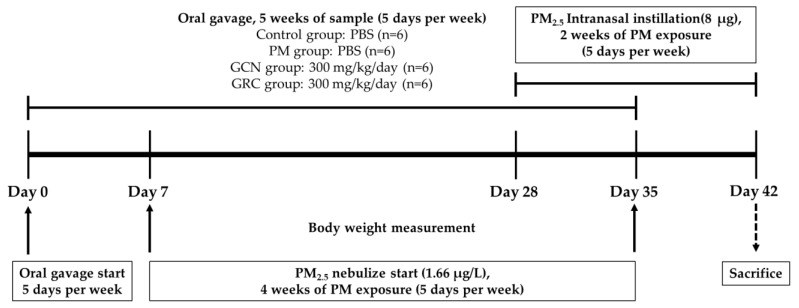
Schematic of the murine experimental schedule. Mice were exposed to PM (1.66 µg/L) through inhalation from day 7 to day 35 and then to intranasal instillation of PM_2.5_ from day 28 to day 42 (5 days/week, *n* = 6). Oral gavage of 100 μL PBS or GRC and GCN (300 mg/kg) was performed daily (5 days/week) for six consecutive weeks in all mice.

**Table 1 ijms-26-01105-t001:** Annual average PM_2.5_ concentrations in Seoul (2006–2011).

Annual Average	2006	2007	2008	2009	2010	2011	AVR
PM_2.5_ concentration (μg/m^3^)	30	30	27	27	28	22	27.3

The amount of particulate matter that Koreans are exposed to during the average period of a day: 27.3 (μg/m^3^) × 500 (mL/breath) × 12 (breath/min) × 60 (min/h) × 24 (h/day) ÷ 10⁶ (mL/m^3^) = 235.87 (μg/day). The annual average PM_2.5_ concentrations and the formula were adopted from Refs. [187,188].

**Table 2 ijms-26-01105-t002:** Semiquantitative scoring of lung injury.

Score	Alveolar Septa	Alveolar Hemorrhage	Intra-Alveolar Fibrin	Intra-Alveolar Infiltrations Per Field
0	All thin and delicate	No hemorrhage	No intra-alveolar fibrin	Less than five intra-alveolar cells
1	Congested alveolar septa in <1/3 of the field	Erythrocytes per alveolus in one to five alveoli	Fibrin strands in less than 1/3 of the field	Five to ten intra-alveolar cells
2	Congested alveolar septa in 1/3–2/3 of the field	At least five erythrocytes per alveolus in five to ten alveoli	Fibrin strands in 1/3–2/3 of the field	Ten to twenty intra-alveolar cells
3	Congested alveolar septa in >2/3 of the field	At least five erythrocytes per alveolus in more than ten alveoli	Fibrin strands in >2/3 of the field	>20 intra-alveolar cells

Source: Ref [189].

## Data Availability

The data are contained within this article.

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
