# Peer review of "Chitosan Nanoparticle-Encapsulated Cordyceps militaris Grown on Germinated Rhynchosia nulubilis Reduces Type II Alveolar Epithelial Cell Apoptosis in PM2.5-Induced Lung Injury"

_ijms, 2025, doi:10.3390/ijms26031105_

Round 1

Reviewer 1 Report

Comments and Suggestions for Authors

- iThenticate report showing 47%; Author should be reduced 

-How can authors claim that the developed formulation was nanosized without measuring the particle size and SEM or TEM analysis? Authors need to perform these analyses to claim the title of the research article.

-4.1. Synthesis of GRC chitosan nanoparticles (GCN) : How much ml chitosan solution was used?

-4.3. In vitro release: add the compositions of all buffers used for release study. How was maintained the sink condition in 5 ml buffer media?

-Add the regression equation used for the calculation of concentrations of GRC.

-Why did not use standard drug to compare the results?

Comments on the Quality of English Language

The English could be improved

Reviewer 2 Report

Comments and Suggestions for Authors

The manuscript describes the investigation of the developed chitosan nanoparticles (CNP) to enhance the limited bioactivity and stability of Cordyceps militaris grown on germinated Rhynchosia nulubilis (GRC) and to effectively deliver it to target tissues. Further in vivo experiments demonstrated that GCN treatment significantly reduced lung injury scores in mice exposed to fine particulate matter (PM2.5) (p?)

The topic and scope fit the special issue “Cellular and Molecular Mechanisms of Acute Lung Injury;” the methods are appropriate and the results support the conclusions.

P values are provided which is good rigor. Abstract misses the last p value.

A large body of literature has been referred to, this is good. However, in Section 3, besides discussion on results and implications, a large fraction is on background and PM2.5 injury mechanism, reviewing the relevant literature. Is this necessary? Please reduce and condense to focus on the results and impact.

Reproducibility can be further improved by providing confidence intervals, alpha values, etc. in section 4.14.

The authors note some relevant trends in structural changes and stabilisations, evidenced through red/blue-shift of the peaks in the FTIR spectra collected. In particular, the electrostatic interactions between relevant groups and formation of H-bonds. The authors can strengthen their arguments through citation of works quantifying these modifications to peak shapes and frequencies, specifically in amid and carbonyl groups. Extending the following sentence and supporting with relevant citation is strongly encouraged:

"... and the appearance of a modified peak at 1639cm-1 in GCN indicate new electrostatic interactions between the -NH3+ groups of CHI and the -COO groups of GRC compounds such as peptides from soybean and C. militaris [88], and suggest tractable changes to the nature of amine, amide and carbonyl groups [new-89]."

[new-89] title Quantitative scale for the extent of conjugation of carbonyl groups:“Carbonylicity” percentage as a chemical driving force.

Fig. 1 caption is missing references.

All of the result figures are overly small, with low resolution and are not readable. Please enlarge.

Fig. 4C has large blank space. Please reduce Y-axis range.

Please double check Line 233-234, which appears conflicting with the conclusions.

Line 256, if this is the result, a reference should not be placed here.

Line 822, equation should be numbered.

Tables 1, 2 are not results, thus the origin should be cited.

Reviewer 3 Report

Comments and Suggestions for Authors

This work describes the development of a chitosan nanoparticle (NP) system encapsulating GRC extract to alleviate the effects of PM exposure. The authors used multiple in vitro experiments to demonstrate the antioxidant and anti-apoptotic effects of GRC-encapsulated chitosan nanoparticles (GCNs). The in vivo results also show positive outcomes of GCNs. Below are several suggestions for consideration.

1.     The resolution of most figures is insufficient and should be improved.

2.     A clear timeline of the in vivo experiment is necessary. The dosing sequence and timing of PM and GCN administration need clarification.

3.     The main statement in the Figure 4 caption appears incomplete or only partly correlated with its contents (lung injury evaluation and toxicity effects of GRC and GCN). The caption should more directly reflect what the figure is illustrating.

4.     The use of a cellular uptake experiment to demonstrate enhanced bioavailability is limited, as in vitro conditions differ significantly from the GI tract. More direct evidence of oral bioavailability enhancement would strengthen the argument that chitosan NPs can facilitate GRC absorption.

5.     Including tissue analyses of the GI tract and the biodistribution of GCN in major organs might offer additional insights and further support.

6.     I appreciate the summary of Figure 9. It is a good summary of possible pathways.

Round 2

Reviewer 1 Report

Comments and Suggestions for Authors

Accepted for publication 

Comments on the Quality of English Language

No comments

Reviewer 3 Report

Comments and Suggestions for Authors

The authors have finely revised the article and provide sufficient information to prove their ideal. I think this work can be published in this journal.